# SELF-SUPERVISED LEARNING OF APPLIANCE USAGE

**Chen-Yu Hsu, Abbas Zeitoun, Guang-He Lee, Dina Katabi & Tommi Jaakkola**
Computer Science and Artificial Intelligence Lab
Massachusetts Institute of Technology
Cambridge, MA 02139, USA
`{cyhsu,zeitoun,guanghe,dk}@mit.edu, tommi@csail.mit.edu`

## ABSTRACT

Learning home appliance usage is important for understanding people's activities and optimizing energy consumption. The problem is modeled as an event detection task, where the objective is to learn when a user turns an appliance on, and which appliance it is (microwave, hair dryer, etc.). Ideally, we would like to solve the problem in an unsupervised way so that the method can be applied to new homes and new appliances without any labels. To this end, we introduce a new deep learning model that takes input from two home sensors: 1) a smart electricity meter that outputs the total energy consumed by the home as a function of time, and 2) a motion sensor that outputs the locations of the residents over time. The model learns the distribution of the residents' locations conditioned on the home energy signal. We show that this cross-modal prediction task allows us to detect when a particular appliance is used, and the location of the appliance in the home, all in a self-supervised manner, without any labeled data.

## 1 INTRODUCTION

Learning home appliance usage patterns is useful for understanding user habits and optimizing electricity consumption. For example, knowing when a person uses their microwave, stove, oven, coffee machine or toaster provides information about their eating patterns. Similarly, understanding when they use their TV, air-conditioner, or washer and dryer provides knowledge of their behavior and habits. Such information can be used to encourage energy saving by optimizing appliance usage (Armel et al., 2013), to track the wellbeing of elderly living alone (Donini et al., 2013; Debes et al., 2016), or to provide users with behavioral analytics (Zhou & Yang, 2016; Zipperer et al., 2013). This data is also useful for various businesses such as home insurance companies interested in assessing accident risks and utility companies interested in optimizing energy efficiency (Armel et al., 2013).

The problem can be modeled as event detection – i.e., given the total energy consumed by the house as a function of time, we want to detect when various appliances are turned on. Past work has looked at analyzing the energy signal from the home utility meter to detect when certain appliances are on.[1] Most solutions, however, assume that the energy pattern for each appliance is unique and known, and use this knowledge to create labeled data for their supervised models. (Kolter et al., 2010; Zhong et al., 2014; 2015; Kelly & Knottenbelt, 2015; Zhang et al., 2018; Bonfigli et al., 2018). Unfortunately, such solutions do not generalize well because the energy pattern of an appliance depends on its brand and can differ from one home to another (Kelly & Knottenbelt, 2015; Bonfigli et al., 2018).[2] The literature also contains some unsupervised methods, but they typically have limited accuracy (Kim et al., 2011; Kolter & Jaakkola, 2012; Johnson & Willsky, 2013; Parson et al., 2014; Wytock & Kolter, 2014; Zhao et al., 2016; Lange & Berges, 2018).

Unsupervised event detection in a data stream is intrinsically challenging because we do not know what patterns to look for. In our task, not only may appliance energy patterns be unknown, but also the energy signal may include many background events unrelated to appliance activation, such as the fridge or HVAC power cycling events.

One way to address this challenge is to consider the self-supervised paradigm. If a different stream of data also observes the events of interest, we can use this second modality to provide self-supervising

---

[1]The utility meter outputs the sum of the energy of all active appliances in a house as a function of time.

[2]For example, a Samsung dishwasher may have a different energy pattern from that of a Kenmore dishwasher.

signals for event detection. To that end, we leverage the availability of new fine-resolution motion sensors which track the locations of people at home (Adib et al., 2015; Joshi et al., 2015; Li et al., 2016; Ghourchian et al., 2017; Hsu et al., 2017b). Such sensors operate as a consumer radar, providing decimeter-level location accuracy. They do not require people to wear sensors on their bodies, can operate through walls, and track people's locations in different rooms.

These location sensors indirectly observe the events of interest. Specifically, they capture the change in user locations as they reach out to an appliance to set it up or turn it on (e.g. put food in a microwave and turn it on). Hence, the output of such sensors can provide a second modality for self-supervision.

But how should one design the model? We cannot directly use location as a label for appliance activation events. People can be next to an appliance but neither activate it nor interact with it. Moreover, we do not assume appliance locations are known a priori. We also cannot use the two modalities to learn a joint representation of the event in a shared space. This is because location and energy are unrelated most of the time and become related only when the event of interest occurs. Furthermore, there are typically multiple residents in the home, making it hard to tell which of them interacted with the appliance.

Our model is based on cross-modal prediction. We train a neural network that, given the home energy at a particular time, predicts the location of the home residents. Our intuition is that appliance activation events have highly predictable locations, typically the location of the appliance. In contrast, background energy events (e.g. power cycling of the fridge) do not lead to predictable locations. Thus, our model uses this learned predictability along with the associated location and energy representation to cluster the events in the energy stream. In addition, we use a mixture distribution to disentangle irrelevant location information of other residents in the home. Interestingly, our model not only learns when each appliance is activated but also discovers the location of that appliance in the home, all without any labeled data.

We summarize the contributions of this paper as follows:

- The paper introduces a new method for self-supervised event detection from weakly related data streams. The method combines neural cross-modal prediction with custom clustering based on the learned predictability and representation. We apply it to the task of detecting appliance usage events using unlabeled data from two sensors in the home: the energy meter, and a location sensor.
- To evaluate our design, we have created the first dataset with concurrent streams of home energy and location data, collected from 4 homes over a period of 7 months. For each home, data was collected for 2 to 4 months. Ground truth measurements are provided via smart plugs connected directly to each appliance.
- Compared to past work on unsupervised learning of appliance usage and a new baseline that leverages the two modalities, our method achieves significant improvements of 67.3% and 51.9% respectively for the average detection F1 score.

We will release our code and dataset to encourage future work on multi-modal models for understanding appliance usage patterns and the underlying user behavior. [3]

## 2 RELATED WORK

**Energy disaggregation** Our work is related to past work on energy disaggregation, which refers to the problem of separating appliance-level energy from a home's total (or aggregate) energy signal. Past work in this domain can be broadly classified into two categories: supervised and unsupervised.

*Supervised methods* assume that the power signatures of individual appliances are available. They use data from individual appliances to obtain models for each appliance power signature, and then use those models to detect appliance events from the aggregate energy signal. Early work learns sparse codes for different appliances (Kolter et al., 2010) or uses a Factorial HMM (FHMM) (Ghahramani & Jordan, 1996) to model each appliance as an HMM (Zhong et al., 2014; 2015). Other work uses matrix factorization approaches to estimate monthly energy breakdowns (Batra et al., 2017; 2018). More recently, neural networks have been used to model appliances (Kelly & Knottenbelt, 2015; Zhang et al., 2018; Jia et al., 2019; Bonfigli et al., 2018), where extracting appliance-level energy is formulated as a de-noising problem. However, supervised solutions typically do not generalize well

---

[3]Project website: http://sapple.csail.mit.edu

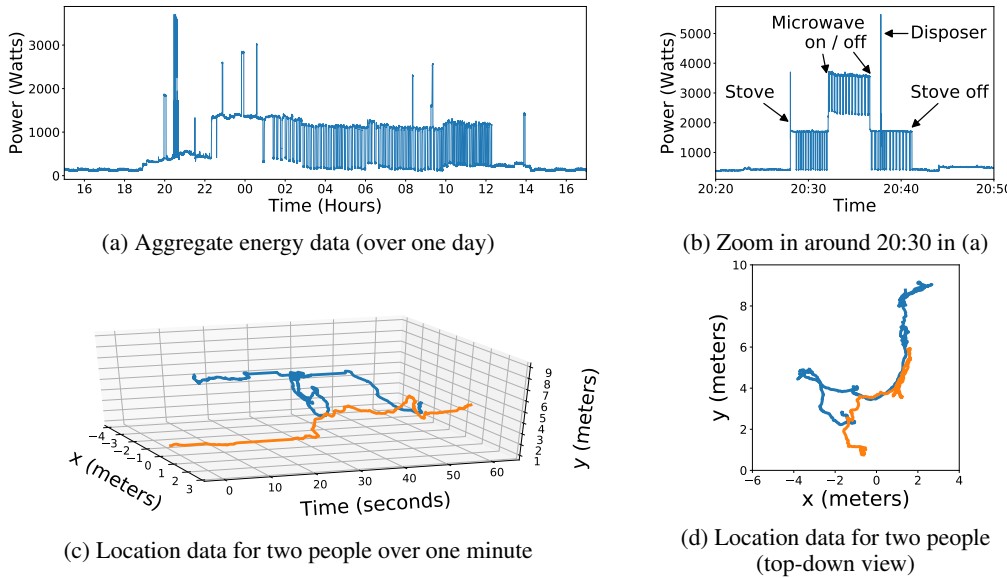

Figure 1: Aggregate energy signal and people's indoor location data

to new homes (Kelly & Knottenbelt, 2015; Bonfigli et al., 2018). This is because two appliances of the same type (e.g. coffee machine) in different homes are often manufactured by different brands, and thus have different power signatures.

*Unsupervised methods* do not assume prior knowledge of appliance signatures; they attempt to learn those signatures from the aggregate energy signal. Early approaches use variants of FHMM, and learn appliance HMMs with Expectation-Maximization (Kim et al., 2011), approximate footprint extraction procedures (Kolter & Jaakkola, 2012), or using expert knowledge to configure prior parameters (Johnson & Willsky, 2013; Parson et al., 2014). Some papers propose using contextual information (such as temperature, hour of the day, and day of the week) (Wytock & Kolter, 2014), or use event-based signal processing methods to cluster appliances (Zhao et al., 2016). More recently, Lange & Berges (2018) proposed using a recurrent neural network as the variational distribution in learning the FHMM. In contrast, our work leverages people's location data as a self-supervising signal. We cluster appliance events through learning the relation between energy events and people's locations, and also learn appliance locations as a by-product.

**Passive location sensing** Motivated by new in-home applications and continuous health monitoring, recent years have witnessed an increasing number of indoor location sensing systems (Adib et al., 2015; Joshi et al., 2015; Li et al., 2016; Ghourchian et al., 2017). They infer people's locations passively by analyzing how people change the surrounding radio signals (e.g. WiFi) and do not require people to wear any sensors. These sensors have been used for various applications including activity recognition (Wang et al., 2014; 2015), sleep monitoring (Zhao et al., 2017; Hsu et al., 2017a), mobility and behavioral sensing (Hsu et al., 2017b; 2019), and health monitoring (Kaltiokallio et al., 2012). In our work, we leverage the availability of such sensors to introduce location data as an additional data modality for learning appliance usage patterns.

**Self-supervised multi-modal learning** Our work is related to a growing body of work on multi-modal learning. Most approaches learn to encode the multi-modal data into a shared space (Gomez et al., 2017; Harwath et al., 2018; Owens & Efros, 2018; Zhao et al., 2018; 2019). In contrast, since our two modalities are mostly unrelated and become related only when an activation event happens, we learn to predict one modality conditioned on the other. Our work is also related to cross-modal prediction (Krishna et al., 2017; Owens et al., 2016; Zhang et al., 2017) but differs from it in an essential way. Past work on cross-modal prediction typically uses the prediction as the target outcome (e.g. output text for video captioning). In contrast, our objective is to discover the hidden appliance activation events. Thus, we design our method to leverage the learned predictability and cross-modal mapping for clustering activation events. Furthermore, we introduce a mixture prediction design to disentangle unrelated information in our predicted modality (location measurements unrelated to energy events).

## 3 PROBLEM FORMULATION

Our goal is to learn appliance activation events in an unsupervised way, using two input streams: home aggregate energy and residents' location data. Figure 1 shows the two data modalities. We describe each of them formally and define appliance "events" below.

**Aggregate energy signal** A household's total energy consumption is measured by a utility meter regularly. This measures the sum of the energy consumed by all appliances at each point in time. We denote the aggregate energy signal by $\boldsymbol{y} = (y_1, y_2, \ldots, y_T)$, where $y_t \in \mathbb{R}_+$. Suppose there are a total of $K$ appliances in a home, and each appliance's energy signal is denoted by $\boldsymbol{x}_k = (x_{1,k}, x_{2,k}, \ldots, x_{T,k})$, where $x_{t,k} \in \mathbb{R}_+$. Only the aggregate energy signal is observed, $y_t = \sum_{k=1}^K x_{t,k} + \epsilon_t$, where $\epsilon_t \sim \mathcal{N}(0, \sigma^2)$ is the background noise.

Figure 1a shows one day of an aggregate energy signal. The base power level shifts constantly throughout the day, depending on the background load (e.g. ceiling lights). Added on top of the base level are the various appliance events. Figure 1b zooms in around 20:30, and shows examples of those events. The stove was turned on around 20:28, and its power continued to cycle between a few levels. While the stove was on, the microwave was also turned on and ran for a few minutes, and the garbage disposer was turned on shortly.

**Indoor location data** We use a single location sensor similar to that in Hsu et al. (2017b) to measure people's indoor locations passively. The sensor sends out radio signals and analyzes the reflections to localize multiple people. Similarly to a regular WiFi router, the sensor has a limited coverage area of up to 40 feet. Suppose there are $P_t$ people in the coverage area at time $t$. The location data is denoted by $\boldsymbol{l}_t = (l_{t,1}, l_{t,2}, \ldots, l_{t,P_t})$, where $l_{t,p} \in \mathbb{R}^2$ is the x-y location of person $p$ at time $t$. We can represent the location data over multiple time frames as $\boldsymbol{l}_{1:T} = (\boldsymbol{l}_1, \boldsymbol{l}_2, \ldots, \boldsymbol{l}_T)$. Figure 1c shows one minute of location data from two people, and Figure 1d shows the data from a top-down view.

**Appliance activation events** When an appliance is turned on, it causes a jump in energy consumption, i.e. a leading edge in the energy signal, as shown in Figure 1b. We call such a pattern an appliance activation event. On the other hand, when an appliance changes its internal state, it can also cause a change in the energy signal as shown in the same figure. We call such a pattern a background event. We are interested in discovering activation events to learn appliance usage patterns. Thus, for each jump in the aggregate signal, we take a time window (of 25 seconds) centered around that jump, and analyze it to detect whether it is an activation event and which appliance it corresponds to.

## 4 MODEL

Our model operates on time windows (25 seconds) centered around jumps in aggregate energy signal, and the corresponding time windows of location data. The model aims to detect appliance activation events by finding windows with highly predictable user locations conditioned on the energy signal.

Figure 2 shows our model. The idea underlying our model is to first learn a representation of appliance event windows that separates the information about appliance type, $z_{t,cat}$, from the shape of the energy signal, $z_{t,cont}$. This is achieved through the appliance energy encoder $E$. We can then use the appliance type encoding to predict the location data through the location predictor $L_e$, which is conditioned on $z_{t,cat}$. Since people's locations have information unrelated to appliance events, the total location predictor is a mixture of $L_e$ and a second module $L_g$ which captures event-independent location information. Below, we describe the design of these modules. More details about the neural network parameters and implementation are discussed in Appendix 8.4.

**Appliance Energy Encoder** Given a window of aggregate energy signal $\boldsymbol{y}_{t:t+w_1} = (y_t, y_{t+1}, \ldots, y_{t+w_1})$,[4] the encoder $E$ encodes the series into an event vector $\boldsymbol{z}_t$. We break the event vector into two parts: a categorical vector $\boldsymbol{z}_{t,cat}$ and a continuous vector $\boldsymbol{z}_{t,cont}$. We aim to capture the appliance type with $\boldsymbol{z}_{t,cat}$ (e.g. microwave vs. dishwasher) and use $\boldsymbol{z}_{t,cont}$ to capture the variability within the signature of the same appliance. A softmax layer is applied to $\boldsymbol{z}_{t,cat}$ to ensure that it is a valid distribution over appliance types. $E$ is parametrized using convolution layers, with one fully connected layer to produce $\boldsymbol{z}_{t,cont}$ and another for $\boldsymbol{z}_{t,cat}$. We denote by $\boldsymbol{\theta}_E$ the parameters of the encoder.

---

[4]We remove the base power level in each window by subtracting the minimum in the window.

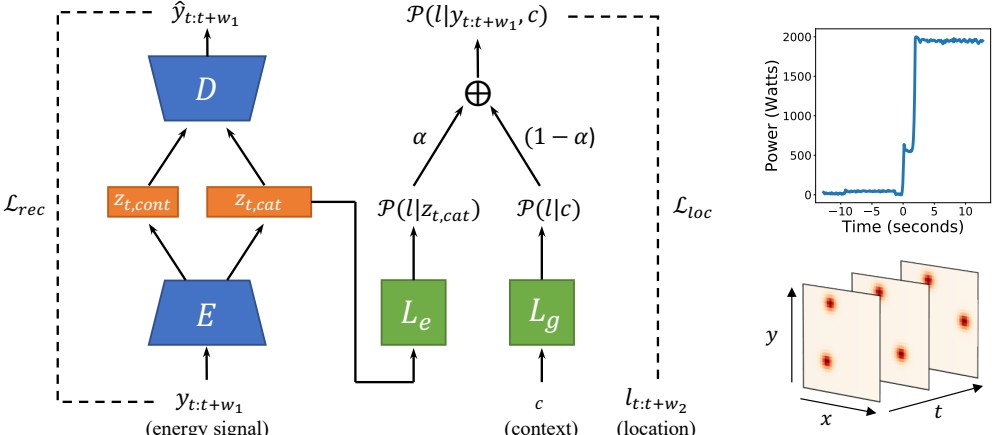

Figure 2: Model architecture. The model learns to encode energy signals into event vectors while learning to predict the concurrent location data. The location predictor $L_e$ is conditioned on the energy event $z_{t,cat}$, and $L_g$ is conditioned on context features. The decoder $D$ takes event vectors and learns to reconstruct the original energy signal.

Figure 3: Total energy signal (top) and location information (bottom) as seen by the model

**Location predictors** We try to predict the location data conditioned on the appliance event, i.e. we predict a window of locations $\boldsymbol{l} = \boldsymbol{l}_{t:t+w_2} = (\boldsymbol{l}_t, \boldsymbol{l}_{t+1}, \ldots, \boldsymbol{l}_{t+w_2})$ centered around the appliance event. We handle multiple people's locations with a mixture model. Specifically, we use $L_e$ to predict locations related to energy events and $L_g$ to handle other locations. The final prediction is a mixture of predictions from $L_e$ and $L_g$:

$$p_{\boldsymbol{\theta}_L}(\boldsymbol{l}|\boldsymbol{y}_{t:t+w_1}, \boldsymbol{c}) = \alpha * p_{\boldsymbol{\theta}_{L_e}}(\boldsymbol{l}|\boldsymbol{z}_{t,cat}) + (1 - \alpha) * p_{\boldsymbol{\theta}_{L_g}}(\boldsymbol{l}|\boldsymbol{c}),$$

where $p_{\boldsymbol{\theta}_{L_e}}(\cdot)$ is parametrized by $L_e$ with parameters $\boldsymbol{\theta}_{L_e}$, $p_{\boldsymbol{\theta}_{L_g}}(\cdot)$ is parametrized by $L_g$ with parameters $\boldsymbol{\theta}_{L_g}$, $\boldsymbol{\theta}_L = \{\boldsymbol{\theta}_{L_e}, \boldsymbol{\theta}_{L_g}\}$, and $\boldsymbol{c}$ includes context features. We use the number of people in the window (reported by the location sensor), the time of day, and the day of the week as the context features. The weight $\alpha$ depends on the number of people in the current window $\alpha = 1/P_t$.

To represent location data, we blur each location measurement with a Gaussian kernel on the x-y plane to create an image, and process the window of locations $\boldsymbol{l}_{t:t+w_2}$ into frames of images (Figure 3). We reuse the notation $\boldsymbol{l} \in \mathbb{R}^{|X| \times |Y| \times |T|}$ to represent frames of location images, where $|X|, |Y|, |T|$ are the number of discretized points on the x-y and time dimensions. By presenting location data as images, we also remove the variable $P_t$ while handling a variable number of people in each frame.

We choose $p_{\boldsymbol{\theta}_{L_e}}(\boldsymbol{l}|\boldsymbol{z}_{t,cat})$ to be a multivariate Gaussian with a diagonal covariance structure: $p_{\boldsymbol{\theta}_{L_e}}(\boldsymbol{l}|\boldsymbol{z}_{t,cat}) \triangleq \mathcal{N}(\boldsymbol{l}; \boldsymbol{\mu}_e, \boldsymbol{\Sigma}_e) = \prod_{x,y,t} \mathcal{N}(\boldsymbol{l}_{x,y,t}; \boldsymbol{\mu}_{x,y,t}, \boldsymbol{\sigma}_{x,y,t}^2)$ where $\boldsymbol{\mu}_e = L_e(\boldsymbol{z}_{t,cat}; \boldsymbol{\theta}_{L_e}) \in \mathbb{R}^{|X| \times |Y| \times |T|}$ and we choose $\boldsymbol{\sigma}_{x,y,t}$ to be a constant. We use 3D deconvolution networks to model $L_e$, which takes $\boldsymbol{z}_{t,cat}$ as input and outputs the means of the location distributions. We model $p_{\boldsymbol{\theta}_{L_g}}(\cdot)$ and $L_g$ in a similar way.

During training, given a window of data $(\boldsymbol{l}, \boldsymbol{y}_{t:t+w_1}, \boldsymbol{c})$, we minimize the negative log likelihood of the mixture distribution in predicting the locations:

$$\mathcal{L}_{loc}(\boldsymbol{\theta}_E, \boldsymbol{\theta}_L) = -\log p_{\boldsymbol{\theta}_L}(\boldsymbol{l}|\boldsymbol{y}_{t:t+w_1}, \boldsymbol{c}).$$

Note that since the gradient flows through $\boldsymbol{z}_{t,cat}$, the likelihood is a function of both $\boldsymbol{\theta}_E$ and $\boldsymbol{\theta}_L$. Hence, the encoder $E$ also learns to encode the energy series based on the concurrent location data.

**Energy Decoder** The decoder $D$ takes both $\boldsymbol{z}_{t,cat}$ and $\boldsymbol{z}_{t,cont}$ and learns to reconstruct the original input energy series by predicting $\hat{\boldsymbol{y}}_{t:t+w_1}$. The decoder $D$ is parametrized using deconvolution layers. We minimize the reconstruction loss during training:

$$\mathcal{L}_{rec}(\boldsymbol{\theta}_E, \boldsymbol{\theta}_D) = ||\hat{\boldsymbol{y}}_{t:t+w_1} - \boldsymbol{y}_{t:t+w_1}||_2$$

The reconstruction loss encourages the encoder $E$ to produce good initial vectors for $L_e$ to predict locations. At the same time, it serves as a regularizer to prevent encoder $E$ from generating meaningless vectors by overfitting location predictions.

**Training** We train all components to jointly optimize the location predictions and energy reconstruction. We minimize the total loss: $\mathcal{L}_{total} = \mathcal{L}_{loc} + \lambda * \mathcal{L}_{rec}$ over all windows of data, where $\lambda$ is a parameter to balance the two terms. [5] The training details are discussed in Appendix 8.4.

## 4.1 CLUSTERING APPLIANCE EVENTS WITH CROSS-MODAL PREDICTIONS

Once the model is trained, we obtain for each window of energy data its appliance event vector $z_{t,cat}$ and its cross-modal location prediction $p_{\theta_{L_e}}(\cdot|z_{t,cat})$. Next, we use these two vectors for clustering. We design a density-based clustering algorithm leveraging the cross-modal relation we learned. Our intuition is that activation events for the same appliance will cluster together since they have the same appliance type and the same location. We omit the *cat* notation below for brevity.

It is typically difficult to cluster in a space learned by a neural encoder because the transformation is highly non-linear and the distance metric is not well-defined. We circumvent this problem by associating the encoded space with a Euclidean space, in which we can easily measure distance. Specifically, for two event vectors $z_1$ and $z_2$, we can measure their distance in the location space using $p_{\theta_{L_e}}(\cdot|z_1)$ and $p_{\theta_{L_e}}(\cdot|z_2)$.

The location prediction $p_{\theta_{L_e}}(\cdot|z_i)$ represents the likelihood of observing any location $l_{x,y,t}$ around the time of the appliance event. We found that for events related to human activities (e.g., turning on a kettle or microwave), $p_{\theta_{L_e}}(\cdot|z_i)$ shows a peak value at the location of the appliance in the x-y space at the time when a person interacted with the appliance. For events not related to human activities (e.g. fridge cycles or random background events), $p_{\theta_{L_e}}(\cdot|z_i)$ has low values and is diffused.

We define the location predictability score (or the confidence of location prediction) as $s(z_i) = \max_{x,y,t} p_{\theta_{L_e}}(l_{x,y,t} = 1|z_i)$, and the location distance $D_{\mathrm{loc}}$ between two events as: $D_{\mathrm{loc}}(z_1, z_2) = ||(x_1^* - x_2^*, y_1^* - y_2^*)||_2$, where $(x_i^*, y_i^*, t_i^*) = \arg\max_{x,y,t} p_{\theta_{L_e}}(l_{x,y,t} = 1|z_i)$.[6] Similarly, the neighborhood distance $D_{\mathrm{nb}}$ between two events is defined as $D_{\mathrm{nb}}(z_1, z_2) = ||z_1 - z_2||_2$.

Our clustering algorithm starts with a $z_i$ with high predictability score $s(z_i)$. It expands the cluster around $z_i$'s local neighborhood in the $z$ space. It stops expanding if a neighbor's location distance $D_{\mathrm{loc}}$ is too far from the cluster center. If all neighbors of the current cluster are visited and none has a small enough $D_{\mathrm{loc}}$, we start a new cluster from another event with high predictability score. The algorithm is described formally in Algorithm 1. We discuss the choice of parameters in Appendix 8.5.

---

**Algorithm 1** Clustering energy events with the learned cross-modal relations

---

**Input:** $\{z_i\}$ and $s(\cdot)$: event vectors and their location predictability scores
 $\eta_s, \eta_{D_{\mathrm{loc}}}, \eta_z$: thresholds for predictability score, location distance, neighborhood distance
 $N_{\min}$: the minimum number of samples to form a valid cluster
**Output:** Clusters of appliance activation events that are associated with a consistent location
 1: **procedure** EL-SCAN($\eta_s, \eta_{D_{\mathrm{loc}}}, \eta_z, N_{min}$)
 2:   $\mathcal{Z} \leftarrow \{z_i|s(z_i) > \eta_s\}, k \leftarrow 0$
 3:   **while** $\mathcal{Z} \neq \emptyset$ **do**
 4:     $z_{\mathrm{seed}} = \arg\max_{\mathcal{Z}} s(z_i)$
 5:     $\mathrm{cluster}_k \leftarrow \{z_{seed}\}, k \leftarrow k + 1$                                    ▷ Start a new cluster
 6:     ExpandCluster($k, z_{seed}, \eta_{D_{\mathrm{loc}}}, \eta_z$)
 7:   **end while**
 8:   **return** clusters with at least $N_{min}$ examples
 9: **end procedure**
10:
11: **function** EXPANDCLUSTER($k, z, \eta_{D_{\mathrm{loc}}}, \eta_z$)
12:   $z_{u_k} \leftarrow$ compute current cluster center
13:   $\mathcal{Z}_{nb} \leftarrow \{z_i \in \mathcal{Z}|D_{\mathrm{nb}}(z_i, z) < \eta_z$ and $D_{\mathrm{loc}}(z_i, z_{u_k}) < \eta_{D_{\mathrm{loc}}}\}$       ▷ Find valid neighbors
14:   $\mathcal{Z} \leftarrow \mathcal{Z} \setminus \mathcal{Z}_{nb}$
15:   $\mathrm{cluster}_k \leftarrow \mathrm{cluster}_k \cup \mathcal{Z}_{nb}$
16:   Repeat ExpandCluster(.) for all $z_i$ in $\mathcal{Z}_{nb}$
17: **end function**

---

[5] We choose $\lambda$ to be 0.1 in our experiments to put more emphasis on the location prediction.
[6] In the implementation, we compute the location predictability score as $\max_{x,y,t} \mu_{x,y,t}$ for simplicity.

## 5 DATASET

We collected concurrent streams of aggregate energy signal and location data from 4 homes over 7 months. [7] We use this dataset for our evaluation. To obtain ground truth labels of appliance events, we deployed programmable smart plugs on the power outlet associated with each appliance. Since not all appliances can be measured by a smart plug (e.g. some appliances do not have accessible power outlets), we also developed a labeling tool for manual labeling. The tool allows labelers to label appliance events from the aggregate energy signal, with the help of smart plug data and information collected from the home residents. The choice of sensors and their sampling rates are detailed in Appendix 8.1.

## 6 RESULTS

We evaluate our model and clustering algorithm on unsupervised appliance activation event detection and their learned appliance locations. We name our approach SAPPLE (Self-supervised APPliance usage LEarning).

### 6.1 UNSUPERVISED APPLIANCE EVENT DETECTION

For appliance event detection, we compare with four baselines. Our method and two baselines have access to location information. *EL-Kmeans* takes both energy and location data as input and directly clusters them using k-means (Arthur & Vassilvitskii, 2007).[8] *E-only-Kmeans* clusters only the energy signal with k-means. Methods with location information pre-filter the events and discard events without any location data, as they are unlikely to be activation events. The other two baselines only take the total energy signal as input: *AFAMAP* (Kolter & Jaakkola, 2012) uses factorial HMM, and *VarBOLT* (Lange & Berges, 2018) uses a recurrent neural network to model aggregate appliance signals. We use publicly available implementations for these methods (implementation, a;b).

We use the same hyper-parameters for the network architecture, training, and clustering algorithm across all homes. As our clustering algorithm is non-parametric, we choose the same number of clusters that it discovers for other methods if possible. For VarBOLT, we report results using 10 clusters, since the training time grows exponentially with the number of clusters and training with more clusters is prohibitively slow. As in past unsupervised work, we report the detection F1 scores based on the best cluster assignments with the ground truth appliances.

Table 1 shows that SAPPLE has an average detection F1 score of 72.8%, outperforming other baselines ranging from 4.0% to 20.9%. As reported by Bonfigli et al. (2018), AFAMAP performs better when appliance-level data is available for training the HMMs. In the unsupervised setting, however, its footprint extraction procedure does not always produce meaningful HMMs for individual appliances (Bonfigli et al., 2018; Beckel et al., 2014), causing degraded performance. VarBOLT's training objective focuses on explaining the total amount of energy in a home. Thus, it often uses multiple components (clusters) to model appliances that are on for a long period (e.g. fridge, heater, and dryer/washer). These types of appliances generate many background events, making the algorithm focus less on activation events of other appliances.

Comparing our method with baselines that also have location information (E-only-Kmeans and EL-Kmeans), our approach still outperforms them significantly. E-only-Kmeans performs better than AFAMAP and VarBOLT, showing that the presence of location data is highly related to activation events. However, naively using location data for clustering does not improve the results by much, as EL-Kmeans performs only slightly better than E-only-Kmeans. This is because not all location data is related to appliance events and vice versa. Our approach "cleans up" the data by learning the relation between the two modalities and discovers clusters with strong cross-modal predictability.

Table 2 shows a break down of our results for different appliances.

### 6.2 ABLATION STUDY

We perform an ablation study to show our results are contributed by all components in our method. As shown in Table 3, we compare our clustering algorithm (Method 1) with a different algorithm that concatenates the learned multi-modal embeddings ($z_{t,cat}$ and $p_{\theta_{L_e}}(\cdot|z_{t,cat})$) and directly clusters

---

[7]The data collection is approved by our institutional review board (IRB).

[8]For each window of data, *EL-Kmeans* concatenates the energy signal, the frames of location images (flattened as a 1-d vector), and the context vector to create the feature vectors clustered by k-means.

Table 1: Unsupervised Appliance Event Detection. Averaged F1 scores (%) of all appliances.

| | $N_{\text{appliances}}$ | Methods w/ location information | | | Methods w/o location | |
| --- | --- | --- | --- | --- | --- | --- |
| | | **SAPPLE** | EL-Kmeans | E-only-Kmeans | AFAMAP | VarBOLT |
| Home 1 | 8 | **82.3** | 26.9 | 10.2 | 6.0 | 4.4 |
| Home 2 | 8 | **69.1** | 19.5 | 19.8 | 5.5 | 3.7 |
| Home 3 | 6 | **76.2** | 16.5 | 15.1 | 4.1 | 3.3 |
| Home 4 | 6 | **63.6** | 20.5 | 20.3 | 6.5 | 4.5 |
| Average | - | **72.8** | 20.9 | 16.4 | 5.5 | 4.0 |

Table 2: Unsupervised Appliance Event Detection.
Our method's F1 score (%) for each appliance.

| | Home 1 | Home 2 | Home 3 | Home 4 |
| --- | --- | --- | --- | --- |
| Kettle | 91.9 | - | - | 98.6 |
| Hair dryer | 88.0 / 98.3 | - | - | 1.1 |
| Coffee machine | 96.1 | 75.6 | 90.4 | - |
| Microwave | 81.9 | 82.1 | 88.1 | 96.7 |
| Stove-activation | 90.6 | 76.7 | - | 92.7 |
| Disposer | 62.5 | 78.5 | 53.1 | - |
| Toaster | - | 49.1 | 71.1 | - |
| Blender | - | 6.2 | - | - |
| Dryer | - | 88.1 | - | - |
| Iron | - | - | 71.1 | - |
| Rice Cooker | - | - | - | 0.0 |
| Others | 49.4 | 96.1 | 83.4 | 92.3 |
| Average | 82.3 | 69.1 | 76.2 | 63.6 |

Table 3: Ablation study.

| | Methods | Avg. F1 (%) |
| --- | --- | --- |
| 1 | **SAPPLE** | **72.8** |
| 2 | Learned embeddings + K-means | 58.8 |
| 3 | Remove $L_g$ | 68.5 |
| 4 | Remove $L_e$ & $L_g$ | 27.6 |

them with k-means (Method 2). Our clustering algorithm is more effective than directly clustering the multi-modal embeddings, providing an improvement of 14.0% in the average F1 score. This is because our clustering algorithm treats the two modalities differently. For location predictions, we can leverage our understanding of physical distance to set cluster boundaries. For the energy embedding, since it is a non-linear mapping with no clear distance metric, our algorithm iteratively groups together events in the embedding neighborhood that have approximately the same locations.

Apart from our clustering algorithm, we evaluate the benefits of our mixture component $L_g$ by experimenting with removing $L_g$ from the model, which reduces the F1 score by 4.3% (Method 1 vs 3). This shows the importance of having $L_g$ extract background motion to allow the location predictor $L_e$ to focus on modeling the person who interacts with the appliance.

We also consider removing both $L_g$ and $L_e$, and clustering the input based only on the energy embedding $z_{t,cat}$ since there is no learned location predictions. The results shown under Method 4 demonstrate the importance of the location embedding generated by the combination of $L_g$ and $L_e$.

### 6.3 LEARNED APPLIANCE LOCATIONS

Our model also learns the locations where people interacted with appliances, which are typically close to the appliances' physical locations (we discuss remotely activated appliances in Appendix 8.6). For each appliance event, we take the location predicted by $L_e$ with the highest predictability score, and compare that with the ground truth appliance location measured by a laser meter. The average location prediction error is 0.65 meters with a standard deviation of 0.17 meters across homes. The errors are mostly due to location offsets between the person and the appliance. Figure 4a shows the location predictions and their ground truth of several appliance events in Home 1. The corresponding energy signals are shown in Figure 4b - Figure 4e.

The location information also helps disambiguate appliances with similar energy signals. For example, although the hair dryer and kettle (Figure 4d and Figure 4e) have very similar energy signatures, their different locations (green and orange in Figure 4a) guide the model to encode their events differently.

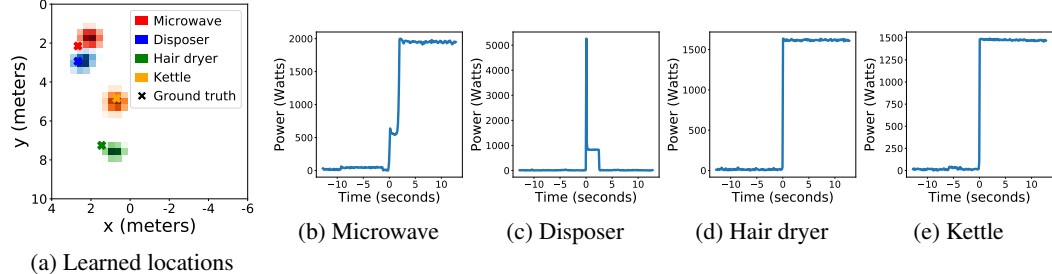

(a) Learned locations    (b) Microwave    (c) Disposer    (d) Hair dryer    (e) Kettle

Figure 4: Energy signals of discovered activation events and their learned locations from $L_e$.

## 6.4 LOCATION PREDICTIONS OF $L_e$ VS $L_g$

We visualize the location predictions from the event-related predictor $L_e$ and the event-independent predictor $L_g$ to illustrate how they handle scenarios with multiple people. Figure 5 shows an example of how the mixture design handles the two types of locations. Since $L_e$ is conditioned on energy events, it naturally learns to predict locations related to appliance events. In this case, the location of the hair dryer is predicted by $L_e$ (Figure 5b). On the other hand, $L_g$ predicts the typical locations people tend to stay (e.g., the couch in Figure 5c) based on the context. Having $L_g$ to explain the other locations helps $L_e$ focus on learning the event-related locations.

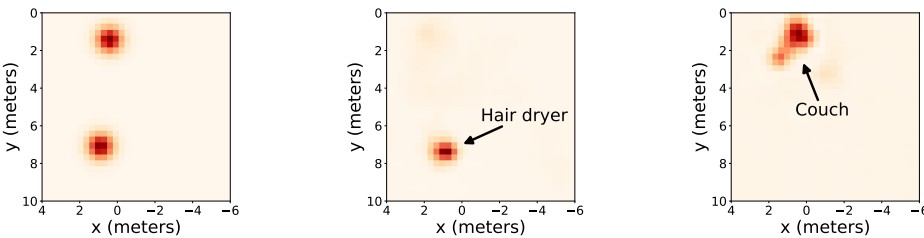

(a) Observed locations (two people)  (b) $L_e$'s prediction (event-related)  (c) $L_g$'s prediction (other locations)

Figure 5: Observed locations and predictions of $L_e$ and $L_g$ at a given time for a hair dryer event.

## 6.5 CONTEXTUAL LOCATION INFORMATION AND CLUSTER VISUALIZATIONS

In Appendix 8.2, we discuss emerging contextual relations between indoor locations through learning cross-modal predictions. We also visualize the learned event vectors to shed light on the design rationales behind our clustering algorithm in Appendix 8.3.

## 7 CONCLUSION

We introduced a self-supervised solution for learning appliance usage patterns in homes. We infer appliance usage by learning from data streams of two modalities: the total energy consumed by the home and the residents' location data. Our approach improves on unsupervised appliance event detection significantly, and learns appliance locations and usage patterns without any supervision. [9]

## ACKNOWLEDGMENTS

The authors thank the members of NETMIT at MIT and the reviewers for their feedback and helpful comments. We thank the participants in our study for facilitating the sensor deployments in their homes. We also thank the various companies sponsoring the MIT Center for Wireless Networks and Mobile Computing.

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

## 8 APPENDIX

### 8.1 SENSORS DETAILS

In this section, we describe details of the sensors used in our dataset collection.

**Aggregate energy signal**    For flexible data collection, we install a sensor (emonPi) at the main circuit breaker in each house as a proxy for the utility meter. We programmed the sensor to collect the raw aggregate energy signal at 1.2 kHz. We down-sampled the data to 10 Hz for our problem to emulate the achievable data rate from a utility meter hardware (Armel et al., 2013).

**Location data**    The wireless location sensor is built on a design similar to Hsu et al. (2017b). It is a single stand-alone sensor that hangs on the wall, and passively collects multiple people's locations with decimeter-level accuracy. We down-sampled the location streams to 1 Hz.

**Appliance-level data (for ground truth labeling)**    We use TP-Link smart plugs (TP-link) with energy monitoring features for collecting appliance-level data. We wrote custom software using available APIs to collect appliance energy signals at 1 Hz. For appliances that cannot be connected to a smart plug, we asked the residents to write down appliance usage times to help with manual labeling.

### 8.2 CONTEXTUAL LOCATION INFORMATION VIA LEARNED APPLIANCE EVENTS

By analyzing the location predictions of $L_e$ conditioned on different appliance events, we also discover interesting contextual relations between different indoor locations. Figure 6 visualizes the location predictions at different frames around a kettle event. We plot the per-frame location predictability score (or prediction confidence) over time in Figure 6a. The score peaks around $t = 0$s, the time of the event. This is because when people turn on a kettle, they may approach it from different locations, but the location when they push the button is consistent and can be predicted confidently. As a result, the prediction at $t = 0$s correctly shows the kettle's location (Figure 6d).

Interestingly, a smaller peak of predictability score shows at $t = -10$s in Figure 6a. If we look at the location prediction from $t = -10$s to $t = 0$s (Figure 6b - Figure 6d), we see how the prediction moves from the sink to the kettle [10]. This is because people often fill water at the sink before starting the kettle. Through learning the cross-modal relation, contextual information among locations also emerges as different appliance events are discovered.

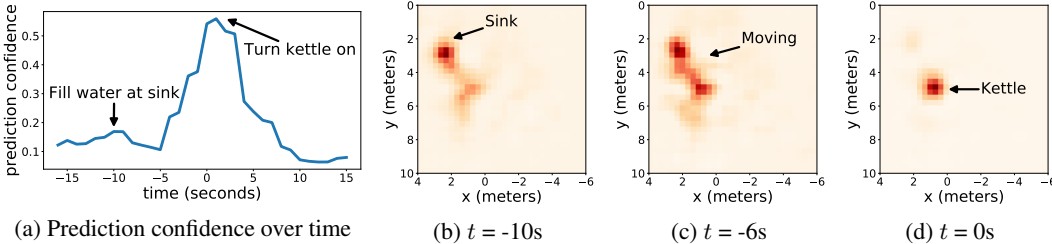

(a) Prediction confidence over time    (b) $t$ = -10s    (c) $t$ = -6s    (d) $t = 0$s

Figure 6: Visualizing location predictions at different times conditioned on a kettle event.

### 8.3 VISUALIZATION OF THE LEARNED EVENT VECTORS AND LOCATION PREDICTABILITY

To illustrate what the model learns and the design rationales behind our clustering algorithm, we visualize the space of the learned event vectors $z_{t,cat}$ and their location predictability score $s(z)$. Figure 7 shows the t-SNE (Maaten & Hinton, 2008) visualization of the event vectors on a 2-dimensional space. We color coded the events with three metrics: location predictability scores (Figure 7a), cluster ID discovered by our algorithm (Figure 7b), and ground truth label (Figure 7c). The predictability score depends on how strongly an appliance event co-occurred with a particular

---

[10]We normalize each image to better visualize locations with lower prediction confidence.

location. As shown in Figure 7a, most appliances related to human activities have high predictability scores (e.g., kettle, hairdryer, microwave, coffee machine, etc). On the other hand, appliances that cycle in the background (e.g., heater) have very low predictability. The stove has many clusters of background events. This is because when the stove is on, it cycles between a few power levels, and the cycle durations depend on the heating levels. Interestingly, we found that stove clusters with higher power levels ("stove-big-cycle") also have high predictability scores, while others with cycling states ("stove-cycle") show low scores. This is likely because people are next to the stove more often when the heating level is high.

We can also see that without clustering using both location predictions and event vectors, it is hard to separate some of the cluster boundaries. Besides, learning to relate energy events to location data enables us to measure the distances of events in a well-defined physical space.

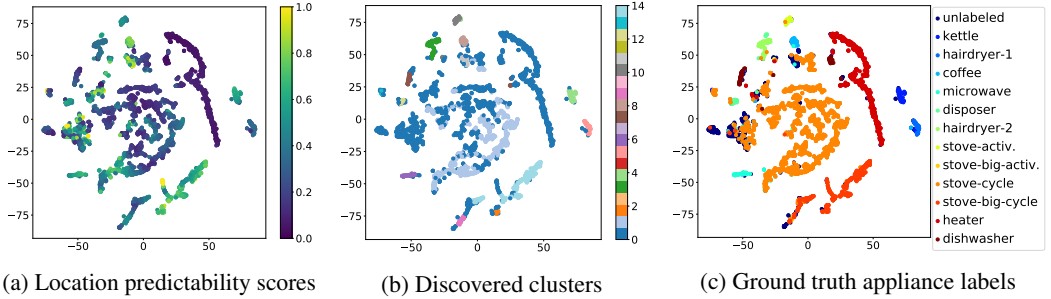

(a) Location predictability scores     (b) Discovered clusters     (c) Ground truth appliance labels

Figure 7: t-SNE visualization of the learned event vectors colored by (a) location predictability scores (b) discovered clusters, and (c) ground truth labels.

## 8.4 NETWORK IMPLEMENTATION AND TRAINING DETAILS

In this section, we provide implementation and training details of our neural network model. We use convolution and deconvolution layers for the energy encoder and decoder. Each module has 8 layers with a kernel size of 3 and a stride of 2. We choose the dimensions of $z_{t,cat}$ and $z_{t,cont}$ to be 128 and 3. The location predictors have 5 layers of 3D deconvolution with a kernel size of 3 and a stride of 2 in each dimension. The frames of location images for each time window have 32 $\times$ 32 $\times$ 32 pixels. We discretize the x, y, and time dimensions into 32 points, where the range of the x-y dimensions are 10 meters and the time dimension has 32 seconds. The neural networks are implemented in Tensorflow (Abadi et al., 2016). For training, we use the Adam (Kingma & Ba, 2014) optimizer with a learning rate of 0.001 and a batch size of 64.

## 8.5 CLUSTERING PARAMETERS AND DETAILS

In all experiments, we set $\eta_{D_{loc}} = 0.4$ meters, $\eta_z = 0.03$, $\eta_s = 0.2$, and $N_{min} = 10$. These values are chosen based on physical and computational constraints. The value of $\eta_{D_{loc}}$ is based on the minimum physical separation between two appliances. The value of $\eta_z$ only affects the search space in each iteration, and is chosen to be small for computational efficiency. The minimum predictability score $\eta_s$ is chosen based on a validation set from one of the homes. The value of $N_{min}$ is set to 10 to say that we need the appliance to appear in the data at least 10 times before we trust that it is a real appliance.

## 8.6 LIMITATIONS

We discuss the limitations of our approach in this section. One limitation is that some remotely activated appliances may not have predictable locations. However, from our experience collecting the dataset, the vast majority of the appliances used on a daily basis (Table 2) require human interaction. For example, a person has to put food into a microwave before turning it on, to hold a hair dryer while drying hair, and to push a button to get a coffee machine running. Even for an appliance with a remote controller, as long as the person has a regular place to interact with the appliance from (e.g., always turning the TV on while sitting on the couch), our model can still learn to predict the location of interaction. Another limitation is that our location sensor has a limited coverage area (around 40 feet in radius). This is enough to cover a typical one-bedroom apartment. For a larger house, one could deploy a second sensor, similarly to how a WiFi repeater extends the coverage area.

