# OpenReview forum: "Self-Supervised Learning of Appliance Usage"
_ICLR.cc/2020/Conference — Accept (Poster)_

### Official Review · AnonReviewer3 · 2019-10-20
**Official Blind Review #3**

**Rating:** 6

**Review:**

Authors proposed a multi-modal unsupervised algorithm to uncover the electricity usage of different appliances in a home. The detection of appliance was done by using both combined electricity consumption data and user location data from sensors. The unit of detection was set to be a 25-second window centered around any electricity usage spike. Authors used a encoder/decode set up to model two different factors of usage: type of appliance and variety within the same appliance. This part of the model was trained by predicting actual consumption. Then only the type of appliance was used to predict the location of people in the house, which was also factored into appliance related and unrelated factors. Locations are represented as images to avoid complicated modeling of multiple people.

As the final step, a customized clustering algorithm was used to turn appliance related events into clusters that each represent an appliance.

Authors trained the model with real world data collected from 4 homes over a few months each. Smart plugs and some human labeling served as ground truth. The results showed good performance of the proposed unsupervised algorithm in recovering the actual appliances in the homes.

Reviewer acknowledge the difficulty in collecting such data and the careful analysis into data characteristics which led to the algorithm design. Reviewer also found the illustrations (esp Fig 1) quite intuitive for understanding the problem. Proposed algorithm is quite reasonable. The release of code and data could also be helpful to the community.

Reviewer also saw some non-trivial issues that Authors may need to fix or explain:

1) Baseline setup seems to be too weak. Authors didn't explain how EL-KMeans was conducted which is non-trivial. The given citation (Arthur & Vassilvitskii, 2007) is a very generic paper discussing KMeans++, not EL-KMeans. In Reviewer's opinion this needs to be fixed for acceptance.

2) in Section 6.2 Ablation study, Author mentioned "we adopt a density propagation algorithm that only uses local neighborhood distance" but Reviewer didn't find any content related to the algorithm. Is this the cluster algorithm?

Other issues:

1) In table 2,  What happened to Home 4's hair dryer and rice cooker? Their F1 are as low as 1.1.

2)  The data size is quite small, which makes the conclusion less strong.


**Experience Assessment:**

I do not know much about this area.

**Review Assessment: Checking Correctness Of Derivations And Theory:**

I carefully checked the derivations and theory.

**Review Assessment: Checking Correctness Of Experiments:**

I carefully checked the experiments.

**Review Assessment: Thoroughness In Paper Reading:**

I read the paper thoroughly.

---

> ### Author Response · Authors · 2019-11-14
> **Response to Reviewer 3**
>
> Thank you for your thoughtful comments. We are glad that you appreciate our data collection, analysis, and the proposed algorithm. We are also glad that you find our illustrations intuitive and our code and dataset helpful to the community.
>
> [Baseline setup]
> We will revise the text to add more details about the baselines. Specifically, EL-KMeans takes the same input as our method, i.e., windows of energy signal and the corresponding windows of location data. For each window, EL-kmeans concatenates the energy signal, the frames of location images (flattened as a 1-d vector), and the context vector to create the feature vector. EL-KMeans then runs KMeans++ on the windows of feature vectors to output the clustering results. We will clarify the explanation in the paper.
>
> The reviewer might also be asking why our method has a big improvement over EL-KMeans. As shown in our results, methods with location information like EL-KMeans do perform better than baselines without location information, showing that location data is useful for detecting appliance activation events. However, simply concatenating location and energy data and clustering them as done by EL-Kmens is not good enough. This is because location and energy data are unrelated most of the time and become related only when an appliance is turned on. Furthermore, there are typically multiple residents in the home, so it is important to design a method to separate the location of the user interacting with the appliance from the locations of other residents. Our model addresses these issues by learning the cross-modal prediction between the two streams.
>
> [Density propagation algorithm]
> Sorry for not being clearer. This sentence tries to explain the intuition behind our clustering algorithm and why it performs well. The intuition is that an appliance cluster should have a concentrated location -- i.e., all instances in the cluster should have their location in the same area (i.e., high density). Thus the clustering algorithm iteratively expands a cluster by adding events that are in the same energy embedding neighborhood but also have approximately the same locations. As the algorithm expands the cluster, visually it “propagates” the cluster to regions with high location density. We will update that sentence to avoid confusion.
>
> [Results in Home 4]
> In Home 4, both the hairdryer and rice-cooker are used only occasionally. As a result, the model does not see enough instances of these appliances to predict their locations with high enough predictability scores. Consequently, the model does not discover them successfully and includes them in the “not detected” cluster. This cluster includes many background events, and hence the f1 score is low for both appliances in that home.
>
> [Data size]
> Our data size (number of homes and duration) is comparable to the commonly used datasets for this task. Past work including unsupervised (AFAMAP [1], VarBOLT [2]) and supervised (NeuralNILM [3], seq2point [4], dAM [5]) methods evaluated on 1 to 5 homes using datasets like REDD [6] and UK-DALE [7].
>
> Collecting such datasets from actual homes for several months is difficult, as acknowledged by the reviewer. It often faces many deployment and system challenges different from collecting a dataset from online resources (e.g., online images or videos), which limits the data size. We hope that our new dataset can help address this problem in the community by providing additional data from a wider variety of homes.
>
> Additionally, our dataset is the first to include both concurrent streams of home energy and residents’ location data. This creates new opportunities for understanding appliance usage and developing multi-modal solutions.
>
> [1] Kolter & Jaakkola, 2012 (see reference in our paper)
> [2] Lange & Berges, 2018 (see reference in our paper)
> [3] Kelly & Knottenbelt, 2015 (see reference in our paper)
> [4] Zhang et al., 2018 (see reference in our paper)
> [5] Bonfigli et al., 2018 (see reference in our paper)
> [6] Kolter, J. Zico, and Matthew J. Johnson. "REDD: A public data set for energy disaggregation research." Workshop on Data Mining Applications in Sustainability (SIGKDD), 2011.
> [7] Kelly, Jack, and William Knottenbelt. "The UK-DALE dataset, domestic appliance-level electricity demand and whole-house demand from five UK homes." Scientific data 2 (2015): 150007.

---

### Official Review · AnonReviewer2 · 2019-10-27
**Official Blind Review #2**

**Rating:** 3

**Review:**

# Review ICLR20, Self-Supervised Learning of Appliance Usage

This review is for the originally uploaded version of this article. Comments from other reviewers and revisions have deliberately not been taken into account. After publishing this review, this reviewer will participate in the forum discussion and help the authors improve the paper.


## Overall

**Summary**

The authors introduce a new method for classifying which appliance was turned on by looking at the change in total household electricity consumption and tracking people's coarse positions in the house.


**Overall Opinion**

The paper is concise and interesting, but for now, I have to reject it because the same major things aren't clear to me:

- Why are you doing this, to begin with, i.e. "... to optimize energy consumption"?. What does energy consumption have to do with classifying which appliance was turned on? There might be the rare edge case of a fridge door left open accidentally but that can't be the main argument here, can it?
- Now, assuming that there is a benefit to this, why can't we use the smart plugs all around? You already use them to provide ground truth. But I imagine they are cheap and easy enough to distribute at large scale, whereas the location sensor was custom-built by you if I understood correctly and probably has to be calibrated for every house.

Other than that, the paper is pretty good. Here are some minor...

## Specific comments and questions

- I got slightly disappointed when I picked up the paper because I was in a robotics mindset and assumed "Self-supervised learning of appliance usage" was some new way for a robot to analyze appliances and push buttons. I'd add a "human" in the title for clarity, e.g. "Self-supervised Learning of Human Appliance Usage" or "Self-supervised Learning of Appliance Usage from Human Location Data". But that's just my opinion, no pressure on this one.

### Abstract

- Please never repeat sentences from the abstract in the introduction or vice versa, good as they might be.

### Intro & Rel. Work

all good

### Problem Formulation

- Please always provide a proper caption for your figure so that the caption contains enough information for the reader to understand what's going on in the figure without having to search for it in the main text. This applies to Fig.1 and even more so to Fig.2 where I'd recommend explaining in broad strokes the information flow.
- Fig. 1 (a) Why is there higher power usage between 02-06 compared to 14-18 o'clock? Is that a heater as the outside temperatures drop overnight? (c/d) I don't think that's a good way of visualizing this. I don't get anything out of diagram (c) and I'd prefer it if you could encode the time aspect in the diagram (d) as color/alpha. E.g. blue agent's oldest position is almost imperceptibly white, growing bluer and darker as the data points get closer to the final step in the recording.

### Model

- Doesn't the softwax require to know the number of appliances a priori?
- How do you deal with $\alpha$ and $P_t$ when a person is at the edge of the visible space? Does this cause problems during training or did this never occur? I could imagine, since the location is noisy, a person could jump in and out of the visible area and cause training instability.

### Datasets

all good

### Results

- What's "Door / leaving" in Table 2? Which appliance is this?
- I think the standard at ICLR is to have the table and figure captions both below the respective objects
- The last 3 sections, 6.3 to 6.5, read a bit rushed and might need a rewrite for clarity.
- Fig. 5 is nice. But shouldn't the signal add up to explain the entire scene? There's a bit of blur next to the couch. Is this a training set vs test set artifact?

### Conclusion

- This needs to be developed a bit more: what are some downsides to your method? Where does it NOT work or what makes it prone to error? And what are future directions based on this work? What's worth researching further or have you solved appliance use detection for good?

---

I'm sure there are some good answers for my main concerns outlined at the beginning and after that, I'd be happy to adjust my rating. The rest are minor issues. Like I mentioned, save for page 8, the paper is well-written and focused.

**Experience Assessment:**

I do not know much about this area.

**Review Assessment: Checking Correctness Of Derivations And Theory:**

I assessed the sensibility of the derivations and theory.

**Review Assessment: Checking Correctness Of Experiments:**

I assessed the sensibility of the experiments.

**Review Assessment: Thoroughness In Paper Reading:**

I read the paper thoroughly.

---

> ### Author Response · Authors · 2019-11-14
> **Response to Reviewer 2 (part 3)**
>
>
> [Other minor issues]
> As the reviewer recommended, we will revise the text to avoid repeating the same sentences in the abstract and introduction and to make it clear from the title that we are referring to appliance usage by a human/user.
>
> Figure and table captions: we thank the reviewer for the suggestion. We made the captions brief due to space limitations. We will revise the text to make the captions more descriptive.
>
> Figure 1 (a): The data was collected from a home with multiple graduate students. The subjects have a late schedule and typically leave home around noon.  14-18  (i.e., 2 pm to 6 pm) is typically the time when no one is at home, and hence has very few appliance events. On the day illustrated in the figure, someone returned home and started preparing dinner around 20 (i.e., 8 pm). The heater was turned on around 22 (10 pm), and continued to produce background events until it went off at 12pm.
>
> Figure 1(c/d): Thanks for the suggestions. We will try to encode time as color/transparency in Figure 1(d).
>
> Softmax: We choose the vector length of z_{t,cat} to be an upper bound on the number of appliances, and aim to learn a sparse dictionary of the appliance types.
>
> Person at the edge of visible space: the location sensor has a relatively large coverage area that is up to ~40 feet away from the sensor. Also, since wireless signals traverse walls, the sensor does not lose people when they leave the room. As a result of large through-wall coverage, and since the appliances of interest are all inside the coverage area (away from its boundary), the training is not impacted when people are at the edge of the coverage area.
>
> Door / leaving event: we apologize for the confusing name. It is a large ceiling light fixture that gets turned on when people answer the door or leave home.
>
> Sec 6.3 and 6.5: Thanks for the feedback. The last page might read a bit rushed because we tried to fit many results into 8 pages. We will revisit it to improve clarity.
>
> Fig 5: We are glad that you like the figure. The reason why L_g’s prediction in Fig 5(c) is blurry is because it is a probability distribution of the likely locations given the context, whereas the locations in Fig.5(a) are sharp because they refer to a specific instance.
>
> Limitations: Thanks for the suggestion. We will include a limitation section. We briefly discuss some limitations below. One limitation is that some remotely activated appliances may not have predictable locations (as discussed in Appendix 8.4). Another is that our location sensor has a limited coverage area (around 40 feet in radius). This is enough to cover a typical one-bedroom apartment or the main kitchen and living room areas of larger apartments. For the larger homes, one could deploy a second sensor, similarly to how a WiFi repeater extends the coverage area.

---

> ### Author Response · Authors · 2019-11-14
> **Response to Reviewer 2 (part 2)**
>
>
> [Why we can’t just use smart plugs all around]
> Unfortunately attaching a smart plug to each appliance incurs a large overhead and has several limitations. This is in fact the reason that past work has focused on methods that use data from the utility meter. We list some of the limitations below.
>
> First, in most modern homes, big kitchen appliances (fridge, microwave, dishwasher, etc.) are embedded into the kitchen cabinets and their plugs are not easily accessible without taking them out of their enclosures. Even when they are not embedded, many large appliances (e.g., stove, washer & dryer) consume very high current and hence their plugs typically have shapes and current levels incompatible with existing smart plugs.
>
> Second, managing a deployment of 10 to 15 smart plugs in one’s home is difficult. Each smart plug has to be continuously connected to a wireless network such as WiFi. However, WiFi coverage can be spotty. This is difficult since a natural location for a smart plug connected to the fridge or the dishwasher is behind that appliance, but the large metallic backs of such appliances significantly attenuate and can block radio signals.
>
> Third, smart plugs are significantly large and bulky. This means that leaving a smart plug plugged into an outlet would no longer allow appliances to be pushed against the wall in front of the outlet. Furthermore, due to their size, smart plugs cover more than one socket when attached to an outlet. As a result, users would need to use extra extension cables to rearrange appliances that would have shared the same outlet without the smart plug.
>
> Finally, kids, house cleaners, or other people who are unaware of the function of those smart plugs may remove them or misplace them. For example, the house cleaner might accidentally disconnect appliances for cleaning and then reconnect them to different smart plugs. As a result, a smart plug that was associated with the coffee machine might now become associated with the kettle, etc. Tracking and ensuring that each smart plug stays operational, connected to WiFi, and associated with its correct appliance for months or years is a significant burden.
>
> In contrast, the location sensor is just one sensor, and it does not require calibration for each home. Although we built our sensor, multiple companies already have similar products. This includes big companies like Texas Instruments and startups like Walabot and Emerald Innovations.

---

> ### Author Response · Authors · 2019-11-14
> **Response to Reviewer 2 (part 1)**
>
> Thank you for your thoughtful comments. We are glad that you find our paper well-written, focused, and interesting. We are also happy that you are open to adjusting the rating and helping us improve the paper. We addressed the issues below.
>
> [Motivation]
> As noted by Reviewer 1, learning appliance usage is useful for improving energy efficiency. There are also other applications, such as health sensing and learning behavioral analytics. We briefly discuss each of these applications below.
>
> For improving energy efficiency, appliance usage information has many consumer, industry, and policy benefits as discussed in [1] (see Table 1 therein for a summary). Various studies have shown that appliance level information can help save 12~20% of energy in residential buildings [1, and citations therein]. There are multiple reasons for this and we explain a few from a utility company’s perspective.
>
> The cost to supply energy changes minute by minute because electricity is difficult to store, and generation cost and energy demands vary frequently [2]. Appliance usage information allows utility companies to better analyze energy usage patterns, and reduce peak demands by providing personalized feedback for different households. Such granular appliance usage information could also improve load forecasting, which is critical for energy purchasing, generation, delivery, and infrastructure planning [1]. These improvements can ultimately lead to more efficient energy markets.
>
> Besides energy efficiency, appliance usage information has applications in health sensing. It provides a passive way to understand user habits and behavior at home. For an elderly person living alone, this information helps caregivers assess whether the person is able to perform basic daily activities such as cooking, eating, washing their clothes, etc. It also alerts the caregiver to changes in the elderly person’s lifestyle such as changes in their eating habits (skipping meals) or changes in their needs for heating or cooling. Furthermore, an elderly person turning on appliances late at night could indicate changes in their sleep habits.
>
> Lastly, generating analytics for how people use appliances is important for multiple businesses. One example is behavior-based home insurance. Similarly to how car insurance companies today reward good driving behavior with lower insurance rates, home insurance companies are interested in appliance usage information for better risk assessments. Detecting and analyzing abnormal energy patterns is helpful for reducing residential accidents such as fires due to fire hazards. Another example is related to e-commerce and consumer companies that are interested in using home appliance information to provide more relevant recommendations. For example, people who use the stove for cooking every day may be interested in cooking-related ads. On the other hand, people who hardly use the stove are unlikely to be good targets of such ads. We will include some of the explanations above in our paper to clarify the motivation.
>
> [1] Armel, K. Carrie, et al. “Is disaggregation the holy grail of energy efficiency? The case of electricity.” Energy Policy 52 (2013): 213-234.
> [2] U.S. Energy Information Administration, U.S. Department of Energy. “Electricity explained - Factors affecting electricity prices.” (2019) https://www.eia.gov/energyexplained/electricity/prices-and-factors-affecting-prices.php

---

### Official Review · AnonReviewer1 · 2019-10-28
**Official Blind Review #1**

**Rating:** 8

**Review:**

This paper proposed a learning algorithm to recover the events of using an appliance and as well as the location of the appliance in a home by using smart electricity meter and a motion sensor installed a home. In the model, the input is a window of electricity energy consumption and context and the output of the model is the location collected by the motion sensor. The appliance activation as the latent variables is learned using a autoencoder architecture.
This work is very interesting and new to the energy efficiency community and non-intrusive load monitoring. This provide an alternative approach to understand how energy was used in a house and so could be used potentially for energy efficiency and malfunction detection of appliances. The data is also new to the community. The paper is well written and the experiments are clear to me. I have some more detailed concerns:
1) One thing is surprising to me: in the autoencoder, you want to learning the latent variables {Z_t, Z_{t,cat}}. It is surprising that for example Z_{t,cat} was exactly the appliance category. I suppose it was aided by the location information presented in P(l|y,c)? I had an experience training an autoencoder for energy disaggregation, but it never worked well because the latent variables Z could be arbitrary.
2) It would be more readable if you the model was provided with more details, for example, the detailed model for L_e(Z_{t,cat};\theta_{L_e}). Although p_{\theta_{L_g}} and L_g were defined in a similar way, they should be explicitly given.


**Experience Assessment:**

I have published in this field for several years.

**Review Assessment: Checking Correctness Of Derivations And Theory:**

I carefully checked the derivations and theory.

**Review Assessment: Checking Correctness Of Experiments:**

I carefully checked the experiments.

**Review Assessment: Thoroughness In Paper Reading:**

I read the paper thoroughly.

---

> ### Author Response · Authors · 2019-11-14
> **Response to Reviewer 1**
>
> Thank you for your thoughtful comments. We are glad that you found the work and dataset interesting and new to the community, and the paper well written and clear.
>
> [Latent variable z_{t,cat}]
> Yes, you are right that location prediction is important in learning a meaningful latent variable z_{t,cat}. Since our model is trained end-to-end, the encoding of z_{t,cat} is guided by location data, allowing the encoder, E, to produce meaningful latent vectors.
>
> [Model details]
> We agree with the reviewer that having more model details in the main paper will improve clarity. Due to the page limit, we discussed our model implementation details in Appendix 8.4, including the network architectures, parameters, and training details. Both location predictors use 5 layers of 3D deconvolution to model the location images. We will try to fit more model details into the main paper.

---

### Author Response · Authors · 2019-11-14
**Minor updates to the paper**

Dear reviewers,

Thank you again for the thoughtful comments. We made minor changes to the paper (colored in blue) to address some of the issues. We clarified EL-kmeans and the sentence explaining our clustering algorithm. We also improved the figure caption, strengthened our motivation, and added clearer pointers to model details and discussions in the appendix.

---

### Decision · Program_Chairs · 2019-12-19

**Decision:**

Accept (Poster)

**Comment:**

Authors proposed a multi-modal unsupervised algorithm to uncover the electricity usage of different appliances in a home. The detection of appliance was done by using both combined electricity consumption data and user location data from sensors. The unit of detection was set to be a 25-second window centered around any electricity usage spike. Authors used a encoder/decode set up to model two different factors of usage: type of appliance and variety within the same appliance. This part of the model was trained by predicting actual consumption. Then only the type of appliance was used to predict the location of people in the house, which was also factored into appliance related and unrelated factors. Locations are represented as images to avoid complicated modeling of multiple people.

The reviewers were satisfied with the discussion after the authors, and therefore believe this work is of general interest to the ICLR community.